# The *rosetteless* gene controls development in the choanoflagellate *S. rosetta*

**Tera C Levin, Allison J Greaney, Laura Wetzel, Nicole King\***

Department of Molecular and Cell Biology, Howard Hughes Medical Institute, University of California, Berkeley, Berkeley, United States

**Abstract** The origin of animal multicellularity may be reconstructed by comparing animals with one of their closest living relatives, the choanoflagellate *Salpingoeca rosetta*. Just as animals develop from a single cell–the zygote–multicellular rosettes of *S. rosetta* develop from a founding cell. To investigate rosette development, we established forward genetics in *S. rosetta*. We find that the rosette defect of one mutant, named Rosetteless, maps to a predicted C-type lectin, a class of signaling and adhesion genes required for the development and innate immunity in animals. Rosetteless protein is essential for rosette development and forms an extracellular layer that coats and connects the basal poles of each cell in rosettes. This study provides the first link between genotype and phenotype in choanoflagellates and raises the possibility that a protein with C-type lectin-like domains regulated development in the last common ancestor of choanoflagellates and animals.

**\*For correspondence:** nking@berkeley.edu

**Competing interests:** The authors declare that no competing interests exist.

**Reviewing editor**: Alejandro Sánchez Alvarado, Howard Hughes Medical Institute, Stowers Institute for Medical Research, United States

## Introduction

The molecular mechanisms underlying animal multicellularity evolved, in part, through the modification of ancient adhesion and signaling pathways found in the unicellular and colonial progenitors of animals. The evolution of the animal molecular toolkit may be reconstructed through the study of the choanoflagellates, the closest living relatives of animals (*Lang et al., 2002*; *Carr et al., 2008*; *Ruiz-Trillo et al., 2008*; *Philippe et al., 2009*; *Paps et al., 2012*). For example, despite the fact that choanoflagellates are not animals, they express diverse genes required for animal multicellularity, including C-type lectins, cadherins, and tyrosine kinases (*Abedin and King, 2008*; *King et al., 2008*; *Manning et al., 2008*; *Nichols et al., 2012*; *Suga et al., 2012*; *Fairclough et al., 2013*), demonstrating that these genes predate the origin of animals. In addition, the architecture of choanoflagellate cells is conserved with animals and helps to illuminate the ancestry of animal cell biology (*Nielsen, 2008*; *Richter and King, 2013*; *Alegado and King, 2014*).

The colony-forming species *Salpingoeca rosetta* promises to be particularly informative about the origins of cell differentiation, intercellular interactions, and multicellular development in animals. Through a process that resembles the earliest stages of embryogenesis in marine invertebrates, single cells of *S. rosetta* undergo serial rounds of cell division to develop into spherical rosette colonies (hereafter, 'rosettes'; *Figure 1*) (*Fairclough et al., 2010*; *Dayel et al., 2011*). Rosette development in choanoflagellates mirrors the transition to multicellularity that is hypothesized to have preceded the origin of animals (*Haeckel, 1874*; *Nielsen, 2008*; *Mikhailov et al., 2009*), although its relationship to animal development is unknown. Recent improvements to the phylogeny of choanoflagellates reveal that colony development may have an ancient origin that extends to the first choanoflagellates and possibly to the last common ancestor of choanoflagellates and animals (*Nitsche et al., 2011*). The possibility that choanoflagellate colony development and animal embryogenesis have a common

**eLife digest** All animals descended from a common ancestor that made the leap from living as a single cell to becoming more complicated, with many cells working together. At first, such a creature would likely have been made from clusters of cells that all had the same function. Eventually, different cells took on different roles, and today animals have many organ systems, each made up of specialized cell types. How the ancestors of animals transitioned from being single celled to multicellular, however, is poorly understood.

It is now possible to reconstruct key steps in the evolution of 'multicellularity' by comparing modern animals with their closest living relatives—the choanoflagellates. These are a group of aquatic microorganisms that can either live as single cells or develop into multicellular colonies. The genes that allow choanoflagellate cells to form colonies are hypothesized to be similar to the genes that the very first animals used to become multicellular.

Now, Levin et al. have studied a choanoflagellate called *S. rosetta*. This species is a good choice, as its genome sequence has been decoded and it is relatively easy to induce *S. rosetta* cells to switch between living on their own or living in spherical colonies called rosettes.

Using a technique known as 'forward genetics', Levin et al. bombarded *S. rosetta* cells with chemicals and X-rays to introduce genetic mutations into the cells. The mutated cells were then grown in conditions that would normally cause *S. rosetta* to form rosette colonies; the cells that continued to live in isolation in these conditions were then studied further, as this meant that mutations had occurred in the genes responsible for colony formation.

Levin et al. identified several mutant *S. rosetta* strains that cannot form rosettes. One of these mutant strains had an altered copy of a gene that Levin et al. named *rosetteless*. The protein produced by the *rosetteless* gene is similar to proteins that connect animal cells to one another in tissues and organs. Normally in rosettes this protein is found outside of the cells, in a secreted structure that joins the cells of the colony together. In the Rosetteless mutants, the protein is often incorrectly made and typically ends up on the wrong part of the cell. Levin et al. further confirmed the importance of the *rosetteless*-encoded protein by creating antibodies that stick to the protein and interfere with its function, thereby blocking rosette formation.

Unraveling the role of the *rosetteless* gene is an important step towards understanding which genes made it possible for single-celled organisms to evolve into complex multicellular animals. Future genetic screens in *S. rosetta* promise to reveal whether *rosetteless* is part of a network of genes and proteins which regulate animal development and could thus illuminate the molecular machinery behind multicellularity in the long-extinct predecessors of animals.

evolutionary history is brought into greater relief when compared with the quite different process of development observed in outgroups of the animal + choanoflagellate clade (e.g., *Capsaspora owczarzaki*; **Sebe-Pedros et al., 2013**), in which isolated cells with different genotypes gather into aggregates.

*S. rosetta* is also notable for its experimental tractability relative to other choanoflagellate species. Importantly, the switch between the *S. rosetta* solitary life style and rosette development is regulated by specific lipids produced by the prey bacterium *Algoriphagus machipongonensis* (**Alegado et al., 2012**). Thus, rosette development can be induced in the laboratory. Moreover, the genome and transcriptome of *S. rosetta* have been sequenced and analyzed, revealing numerous homologs of diverse animal genes, many of which are up-regulated in colonies (**Fairclough et al., 2013**). However, the roles of animal gene homologs in choanoflagellates have not been determined, and there have not been any published reports of successful disruptions to choanoflagellate gene function (including gene deletions, RNA interference, or transgene expression). Indeed, no direct functional links have yet been drawn between genotype and phenotype for any choanoflagellate gene or trait. We recently found that *S. rosetta* can be induced to undergo sex and meiosis, suggesting that it may be amenable to mapping crosses (**Levin and King, 2013**). Therefore, to determine the genetic basis of rosette development and investigate its relationship to animal development, we set out to establish forward genetics in *S. rosetta*.

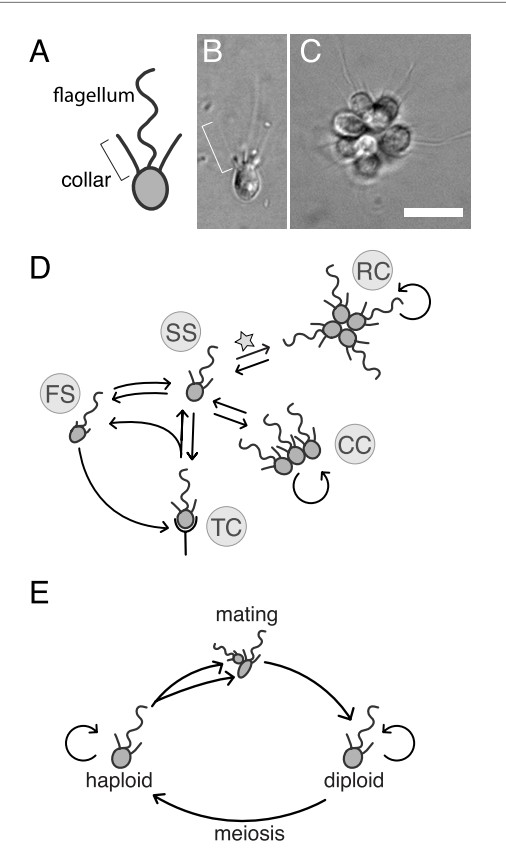

**Figure 1**. *S. rosetta*: an emerging model for studying animal origins and multicellularity. *S. rosetta* cells are polarized, each having a single apical flagellum encircled by a collar of microvilli (bracket), shown in cross-sectional diagram (**A**) and through DIC imaging of a live cell (**B**). In rosette colonies (**C**), each cell is oriented around a central point, with the flagella facing outward. Bacterial prey (~1 μm rods) attach transiently to the collars of some cells prior to ingestion by phagocytosis. Scale bar = 10 μm. (**D**) *S. rosetta* transitions between several morphologically differentiated cell types during its life history: rosette colonies (RC), chain colonies (CC), slow swimmers (SS), fast swimmers (FS), and thecate cells (TC). The transition from slow swimmers to rosette colonies (star) is induced by lipids from the bacterium *Algoriphagus machipongonensis* and can be regulated in the laboratory. (**E**) *S. rosetta* undergoes a sexual cycle in the laboratory. When starved, haploid cultures produce anisogamous gametes that are capable of mating to produce diploids. Diploids undergo meiosis and thereby produce haploids when grown in nutrient-rich media. Haploids and diploids can also reproduce asexually through mitosis.

# Results

## Isolation of mutants with diverse rosette defects

To induce mutations in *S. rosetta*, cultures of haploid cells were exposed either to 0.3% EMS or 6300 rems X-rays, which resulted in a 10% or 40% reduction in cell number, respectively, when averaged across multiple trials (*Figure 3—figure supplement 1A*). We elected to use these relatively light mutagen doses to minimize the number of background mutations in any mutant of interest. After exposing cells to either EMS or X-rays, clonal lines of potential mutants were established by isolating individual cells through limiting dilution (i.e., on average, plating less than one cell/well) into 96-well plates containing rosette-inducing *A. machipongonensis* conditioned media (ACM; *Figure 2*). After 5 to 7 days, each well seeded with a wild-type cell was filled with rosettes, while wells seeded with mutant cells defective in rosette development were expected to produce cultures of solitary cells and/or chain colonies, but few to no rosettes, even in the presence of ACM.

We screened 15,344 clonal cultures for the presence or absence of rosettes (*Figure 2*). Nine mutants with validated rosette defects were isolated ('Materials and methods'), each of which showed a significant reduction in rosette development relative to wild type (*Figure 3A*). The nine rosette defect mutants fell into seven phenotypic classes (classes A–G, *Table 1*; *Figure 3*) based upon their ability to form rosette colonies in the presence of ACM or live *A. machipongonensis*, their swimming behavior as solitary cells, and the morphology of chain colonies produced when grown in the absence of ACM.

Class A consisted of a single mutant, named Rosetteless, that was isolated after EMS treatment. In the presence of either ACM or live *A. machipongonensis*, Rosetteless cells failed entirely to develop into rosettes, but their cell morphology and proliferation were otherwise indistinguishable from wild type (*Figure 3B* and *Figure 3—figure supplement 1B*). Mutants from classes B–D formed some rosettes, but significantly fewer than wild-type strains, and class D exhibited altered spacing and orientation of cells within the rare rosettes that formed (*Figure 3A*, *Figure 3—figure supplement 1C and D*). Rosette development was never observed in mutants from classes E–G (*Figure 3A*, *Figure 3—figure supplement 1C,E*, *Table 1*).

Although *S.rosetta* was originally isolated as a rosette, wild-type cells can produce linear, 'chain' colonies when grown without *A. machipongonensis* (*Figure 1*). Rosettes and chain colonies can be easily distinguished from each other. In addition to the differences in their morphology, the

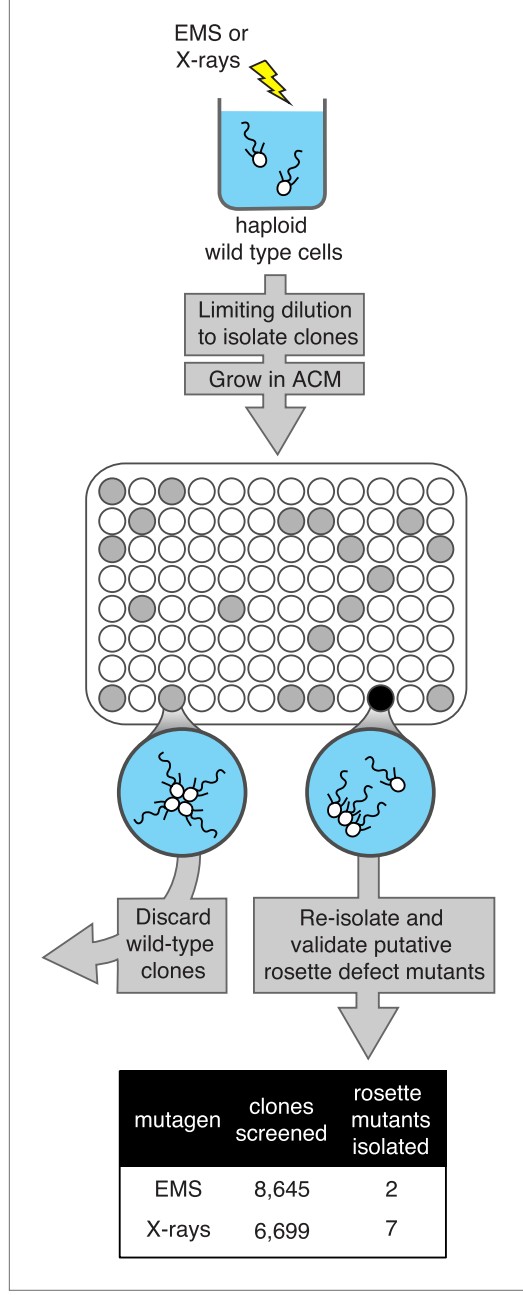

**Figure 2**. A screen for rosette defect mutants in *S. rosetta*. Rosette defect mutants were isolated by exposing *S. rosetta* haploid cells to either EMS or X-rays and then isolating clones in rosette-inducing *Algoriphagus* conditioned media (ACM) prior to visual screening. The use of limiting dilution to isolate clones resulted in many wells with no cells (indicated as white circles). Wells seeded with a wild-type cell (gray circles) produced a culture with abundant rosette colonies, while wells seeded with a rosette defect mutant (black circle) produced a culture with chains or single cells, but few to no chain colonies. Candidate rosette defect mutants were validated through repeated rounds of limiting dilution prior to re-screening in ACM.

connections among cells in rosettes are robust and resistant to mechanical shear, whereas chain colonies are fragile and readily fall apart into individual cells when exposed to shear forces. Nonetheless, chain colonies and rosettes have some similarities, including the presence of fine intercellular bridges connecting neighboring cells and similar transcriptional profiles (*Dayel et al., 2011*; *Fairclough et al., 2013*). We therefore investigated whether the rosette defect mutants had co-occurring defects in their ability to form normal chain colonies with linear morphology. Mutant classes A–C formed apparently wild-type chain colonies. In contrast, classes D–F developed into highly branched chain colonies while cultures of the class G mutant, which rarely formed chains, were instead observed to be predominantly single celled (*Table 1*, *Figure 3—figure supplement 2*).

The ability to isolate mutants with a range of rosette and chain phenotypes demonstrates the potential of forward genetics to illuminate diverse aspects of multicellular development in *S. rosetta*.

## Rosetteless phenotype maps to the gene *EGD82922*

The Rosetteless mutant phenotype was highly penetrant and yet the mutant lacked any other obvious defects (*Figure 3*, *Figures 3—figure supplement 1B and 2*). We thus inferred that the gene(s) disrupted in the Rosetteless mutant might have roles specific to rosette development. Therefore, as we set out to establish methods for mapping mutations in *S. rosetta*, we focused on the Rosetteless mutant. We started by sequencing Rosetteless and two closely related wild-type strains (the parental strain from which Rosetteless was isolated and C2E5, a co-isolated wild-type strain) to identify sequence variants that could serve as genetic markers (*Figure 4—figure supplement 1*, 'Materials and methods'), with the understanding that one or more of the detected sequence variants might ultimately prove to be the causative mutation(s). After filtering the sequence variants by quality, we identified 25,160 potential genetic markers that differed between Rosetteless and the reference genome sequence, only four of which were unique to the Rosetteless genome (*Figure 4—figure supplement 1*).

Our recent discovery of the sexual cycle of *S. rosetta* (*Levin and King, 2013*) suggested that it might be possible to perform a choanoflagellate mapping cross to identify the mutation(s) responsible for the Rosetteless phenotype. To this end, Rosetteless was mated with another, previously sequenced *S. rosetta* strain, Isolate B (*Levin and King, 2013*), that carried 39,451

putative sequence polymorphisms relative to Rosetteless (*Figure 4—figure supplement 2*). Using a combination of serial dilutions and genotyping, we isolated seven outcrossed diploids and established clonal cultures (*Figure 4*, 'Materials and methods'). In the second phase of the mapping cross, the heterozygous diploid cultures were expanded and divided into multiple flasks, rapidly passaged in rich media to induce meiosis, and subjected to another round of serial dilution to generate clonal cultures. Of 442 clonal cultures genotyped, 182 were haploid progeny of the cross, as evidenced by their homozygosity at three microsatellite markers ('Materials and methods').

Genotyping of each haploid isolate at 60 polymorphic sites across the genome revealed that most markers followed Mendel's law of Segregation and Independent Assortment ('Materials and methods', *Figure 4—source data 1*) (*Mendel, 1866*); thus *S. rosetta* inheritance appears to follow the rules of classical genetics. Analysis of the genotyping data in haploids also revealed genetic linkage among some of the markers, allowing us to generate a linkage map containing 27 preliminary linkage groups that represent approximately 70% of the *S. rosetta* genome (*Figure 4—figure supplement 3*, *Figure 4—source data 1*).

Most importantly, the genotype data revealed only one mutation (supercontig 8, position 427,804) that was tightly linked (<0.56 cM) to the rosette defect phenotype. The presence of the mutation was linked to the presence of the Rosetteless phenotype in all examined haploid progeny from the Rosetteless × Isolate B cross (177/177, *Figure 4*). Moreover, the mutation was one of the four validated Rosetteless-specific SNVs and, by disrupting a splice donor in the gene *EGD82922*, was the only one predicted to cause a coding change (*Figure 4—figure supplement 1B*). To investigate whether our variant calling method was too restrictive, we also genotyped the heterozygous diploids for 20 additional putative polymorphisms near the *EGD82922* marker, which were called below our quality threshold, and none proved to be polymorphic in this cross. Therefore, based on the tight linkage between the *EDG82922* mutation and the phenotype, as well as the absence of any other

**Table 1.** Classification of mutant phenotypes

| | | Observed rosette induction* | | Other phenotypes | |
| --- | --- | --- | --- | --- | --- |
| | Mutagen used | ACM | Live bacteria | Swimming† | Chain morphology |
| Wild type | N/A | 86% | 88% | Wild type | Primarily linear |
| Mutant class A | | | | | |
| Rosetteless | EMS | 0 | 0 | Wild type | Primarily linear |
| Mutant class B | | | | | |
| Insensate | X-rays | 0 | 5 | Wild type | Primarily linear |
| Mutant class C | | | | | |
| Slacker | X-rays | 20 | 42 | Wild type | Primarily linear |
| Mutant class D | | | | | |
| Uptight | X-rays | 33 | 56 | Wild type | Branched |
| Mutant class E | | | | | |
| Jumble | EMS | 0 | 0 | Wild type | Branched |
| Branched | X-rays | 0 | 0 | Wild type | Branched |
| Mutant class F | | | | | |
| Seafoam | X-rays | 0 | 0 | Wild type | Large clusters |
| Soapsuds | X-rays | 0 | 0 | Wild type | Large clusters |
| Mutant class G | | | | | |
| Solo | X-rays | 0 | 0 | Slow, shaking | Primarily solitary |

*The percentage of cells in rosettes following induction.
†Swimming phenotypes of single cells.

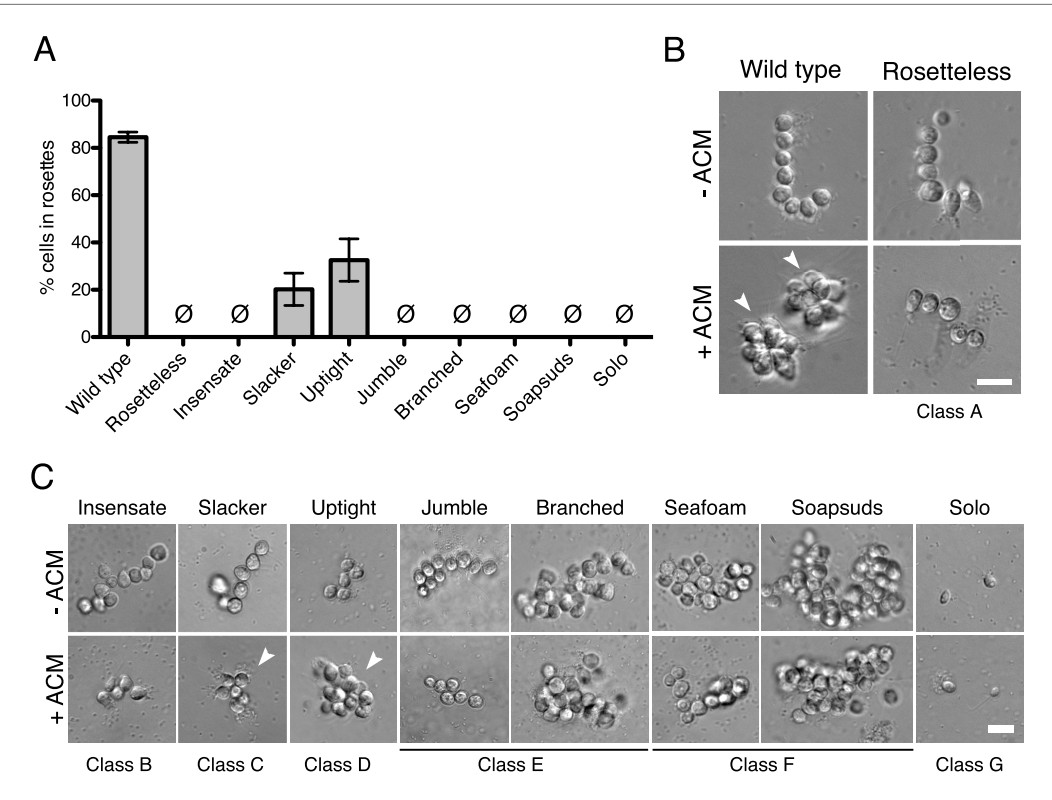

**Figure 3**. Phenotypes of diverse rosette defect mutants. (**A**) Cultures of all nine mutants isolated in this study showed a significantly reduced number of cells in rosettes relative to wild type (one-tailed Mann–Whitney test, p < 0.01). Rosette development was measured as the % of cells in rosettes after 48 hr in 20% ACM, shown as mean ± SEM. Ø indicates mutants in which no rosettes were observed (limit of detection = 0.03%). (**B**) Wild-type *S. rosetta* grown without ACM formed flexible, linear chains or single cells (***Figure 3—figure supplement 2***). When exposed to ACM, wild-type *S. rosetta* cultures produced spherical rosettes (arrowheads). Rosetteless cultures did not form rosettes in ACM, but otherwise appeared in wild type, forming normal chain colonies and proliferating at rates indistinguishable from wild-type *S. rosetta* (***Figures 3—figure supplement 1B,E and 2***). (**C**) Unlike Rosetteless, the remaining eight rosette defect mutants showed additional phenotypic aberrations. Although a small percentage of Slacker and Uptight cells were found in bona fide rosettes (arrowheads), most remained as single cells or chain colonies that were easily disrupted when exposed to shear (***Figure 3—figure supplement 1E***). Seafoam and Soapsuds formed large, disorganized clusters of cells that were easily disrupted when exposed to shear (***Figure 3—figure supplement 1E and 2***) and were thus not rosettes. Scale bars = 10 μm.

The following figure supplements are available for figure 3:

**Figure supplement 1**. Mutagenesis and mutant phenotypes.

**Figure supplement 2**. Chain colony morphologies of diverse mutants.

detected polymorphisms in the 5′ end of supercontig 8, we infer that the candidate splice donor mutation in the gene *EGD82922* is responsible for the rosette defect phenotype. We hereafter refer to *EGD82922* (Genbank accession XP_004995286) as *rosetteless* (*rtls*) and the relevant mutation as *rtls[l1]*.

## A predicted C-type lectin required for rosette development

The *rtls* gene encodes a 119 kDa protein with an N-terminal signal peptide and two C-type lectin-like domains (CTLDs; ***Figure 5A***). CTLD-containing proteins, including the C-type lectins, are found in all animal lineages and play diverse roles, including cell–cell adhesion, cell–extracellular matrix adhesion, cell signaling, and innate immune recognition of pathogens through their binding to carbohydrates,

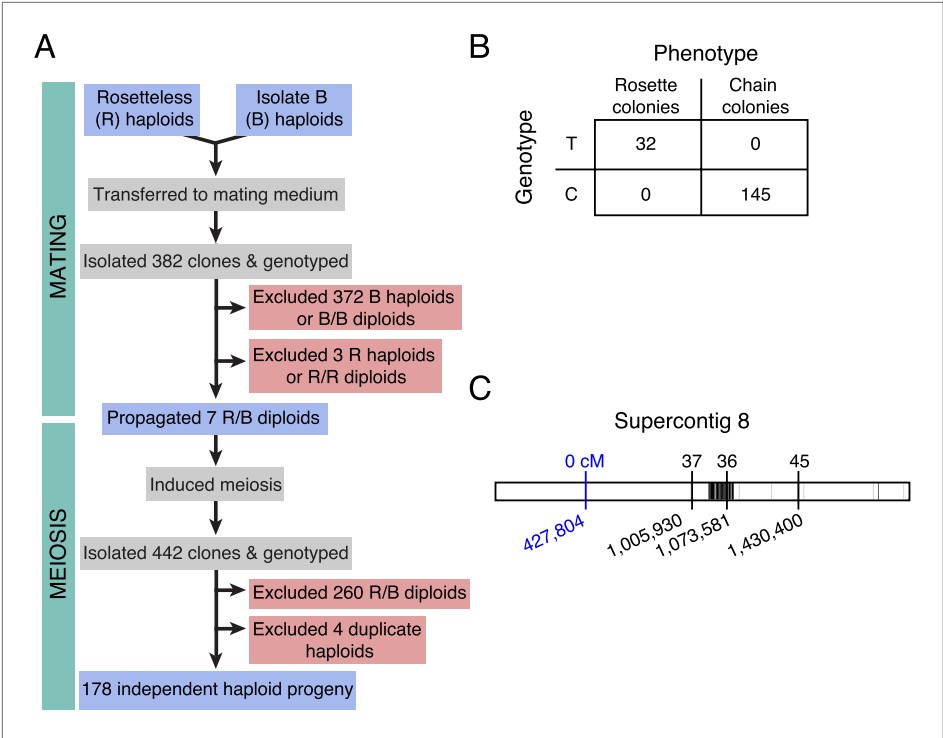

**Figure 4**. Rosetteless maps to *EGD82922*. (**A**) Design of the mapping cross. Rosetteless cells were mixed with Isolate B, an *S. rosetta* culture capable of forming rosettes. Mating was induced by starvation. To isolate the products of outcrossed mating, cells then underwent clonal isolation, and clonal populations were genotyped to identify outcrossed, diploid heterozygotes. These heterozygotes were expanded and induced to undergo meiosis, after which clonal isolation and genotyping were repeated. Haploid progeny of the cross were homozygous at all three markers. (**B**) 2 × 2 contingency table shows that the Rosetteless phenotype was tightly linked to the genotype of the supercontig 8: 427,804 candidate splice donor mutation. (**C**) Map of the supercontig 8 markers. Top numbers show the genetic distance between the markers and the Rosetteless phenotype in centimorgans (cM). Bottom numbers show marker genomic positions on supercontig 8. Black lines within the central bar show all sites of predicted polymorphism (i.e., possible marker positions) between Rosetteless and Isolate B. The blue marker is the *EGD82922* splice donor mutation.

The following source data and figure supplements are available for figure 4:

**Source data 1**. Full genotyping data for all progeny of the Rosetteless x Isolate B cross.

**Figure supplement 1**. Identification of Rosetteless-specific mutations.

**Figure supplement 2**. Map of polymorphisms and markers used in the cross.

**Figure supplement 3**. A linkage map for *S. rosetta*.

proteoglycans, lipids, and other ligands (*Ruoslahti, 1996*; *Cambi et al., 2005*; *Zelensky and Gready, 2005*; *Geijtenbeek and Gringhuis, 2009*; *Švajger et al., 2010*). Similar to CTLDs from animals, the Rtls CTLDs contain four conserved cysteines required for two disulfide bonds, as well as the Glu-Pro-Asn motif (*Figure 5B*) that is required for mannose binding in some C-type lectins (*Drickamer, 1992*). Nonetheless, because the CTLDs of Rtls have not yet been shown to bind sugar moieties, we follow the convention of the field and provisionally refer to Rtls as a C-type lectin-like protein. Rtls also contains several low-complexity regions that each consists of as many as 50–60 consecutive threonines and serines and two high-complexity internal repeat regions (RP1 and RP2) of unknown function near the C-terminus of the protein (*Figure 5A,C*). The serine-threonine-rich regions resemble mucin-like domains found in some animal C-type lectins and are likely sites of O-linked glycosylation (*Carraway*

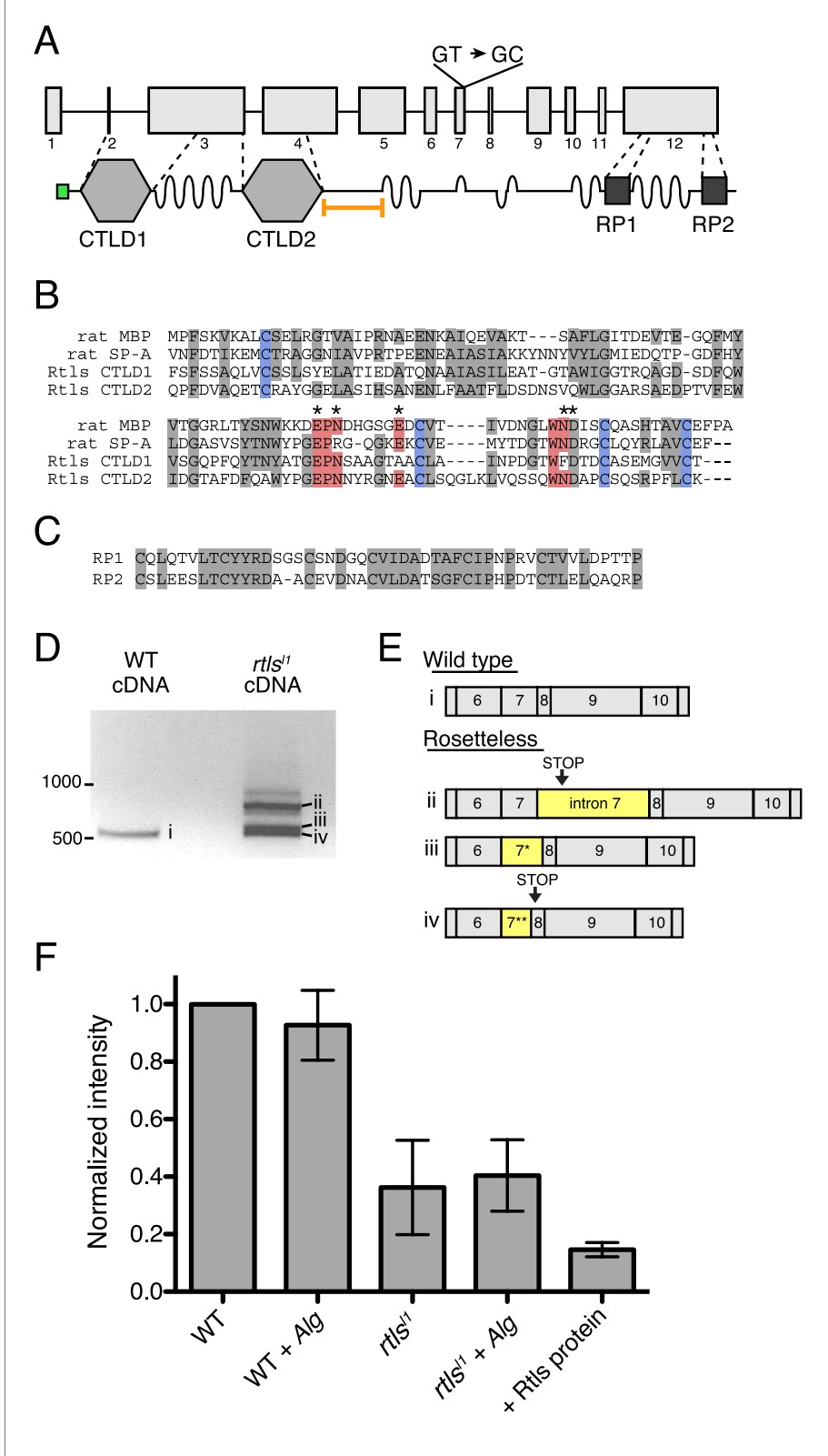

**Figure 5**. Gene structure, domain organization, and expression of *rtls*. (**A**) The *rtls* gene (top) contains 12 exons (numbered) and encodes a protein (bottom) with an amino-terminal signal peptide (green), two C-type lectin-like domains (CTLDs), extended stretches of serines and threonines (wavy lines), and two internal repeats of unknown
*Figure 5. Continued on next page*

*Figure 5. Continued*

function (RP1 and RP2). The *rtls^{l1}* SNV interrupts a splice donor in intron 7 (GT → GC). The epitope used to generate the anti-Rtls antibody is shown (orange bracket). (**B**) An alignment of Rtls CTLDs with CTLDs from rat surfactant protein A (rat SP-A, 1R13_A) and rat mannose-binding protein (rat MBP, 2MSB_A) revealed that residues used in disulfide bonds (blue), mannose-type sugar binding (red), and calcium ion binding (*) are conserved. Other conserved or similar residues are highlighted in gray. (**C**) Alignment of the RP1 and RP2 regions. (**D**) RT-PCR of *rtls* with primers to the exon 5/6 junction and exon 12 showed that wild-type cells produce a single isoform while Rosetteless cells produce diverse splice isoforms. (**E**) Wild-type cDNA yielded the expected splice isoform (i) while Rosetteless mutant cDNA yielded isoforms with: (ii) intron 7 retention or (iii–iv) variants of exon 7 that were longer (*) or shorter (**) than wild type. Isoforms ii and iv contained early stop codons (arrows). (**F**) Semi-quantitative analysis of the fluorescent signal observed in Rtls dot blots, normalized to the intensity of the wild-type culture (WT). Rosetteless mutant cells (*rtls^{l1}*) showed reduced Rtls signal both with and without *A. machipongonensis* (*Alg*) relative to WT (***Figure 5—figure supplement 1C***). Error bars show standard deviation.

The following figure supplements are available for figure 5:

**Figure supplement 1**. Rosetteless splicing and protein levels.

**Figure supplement 2**. The diversity of *S. rosetta* and *M. brevicollis* CTLD-containing proteins.

***and Hull, 1991***; ***Drickamer and Dodd, 1999***). The *rtls^{l1}* mutation, a T-to-C mutation in the predicted splice donor of intron 7, falls 3′ of the sequences encoding the two CTLDs and 5′ of the RP1 and RP2 sequences (***Figure 5A***).

We hypothesized that the exon 7 splice donor mutation in Rosetteless cells might result in defective splicing of *rtls*. To test whether proper *rtls* splicing of exons 7 and 8 can occur in Rosetteless cells, we performed RT-PCR using primers that selectively amplify *rtls* splice isoforms with the predicted exon 7/8 junction and recovered the expected splice isoform from both wild-type and Rosetteless cells (***Figure 5—figure supplement 1A***, ***Supplementary file 1***). However, when using primers that could amplify either the wild-type isoform or variant splice isoforms (a 5′ primer bridging the exon 5/6 junction paired with a 3′ primer in exon 12), we found that *rtls* was spliced as predicted in wild-type cells, but produced multiple, variant splice isoforms in Rosetteless cells. The variant isoforms included one isoform in which intron 7 was retained and two smaller isoforms in which an alternative splice donor either 14 bp upstream or 27 bp downstream of the mutation was used instead (***Figure 5D,E***). Importantly, the wild-type *rtls* isoform was not detected in Rosetteless cells using this assay. For two of the major splice isoforms in Rosetteless cells, the altered splicing led to frame shifts and early stop codons downstream of the mutation, which may either lead to a truncation of the Rtls protein or to degradation of the transcript by nonsense mediated decay in mutant cells (***Lareau et al., 2007***). To investigate endogenous Rtls protein in *S. rosetta* cells, we generated an antibody against residues 438–539, a region of the protein that is unique to Rtls and expected to be present in all wild-type and mutant Rtls isoforms (***Figure 5A***, 'Materials and methods'). Using this antibody, we found that total Rtls protein levels in mutant cells were ~25% that of wild-type cells (***Figure 5F***, ***Figure 5—figure supplement 1B–D***).

The lack of transgenic approaches in choanoflagellates meant that we could not complement the *rtls^{l1}* mutation nor delete the *rtls* gene in wild-type cells. Nonetheless, C-type lectins in animals have been functionally perturbed through the use of blocking antibodies (***Tassaneetrithep et al., 2003***; ***Liu et al., 2014***), and we hypothesized that we could block the function of the extracellular pool of Rtls protein by incubating wild-type cells with an anti-Rtls antibody ('Materials and methods'). Therefore, to test the necessity of Rtls function for rosette development, wild-type *S. rosetta* cultures were incubated with 0–50 μg/ml anti-Rtls antibody during exposure to *A. machipongonensis* bacteria. Treatment with anti-Rtls resulted in significant inhibition of rosette formation relative to negative controls (***Figure 6A*** and ***Figure 6—figure supplement 1A***). Specifically, in cultures treated with 50 μg/ml anti-Rtls during rosette induction, only 10 ± 11% of cells were observed in rosettes (mean ± standard deviation, ***Figure 6A***). In contrast, wild-type *S. rosetta* cultures incubated with an equal volume of rabbit pre-immune serum or an equivalent concentration of normal rabbit IgG or BSA showed normal levels of rosette development, with 91 ± 2%, 90 ± 2%, or 88 ± 1% of cells in rosettes, respectively.

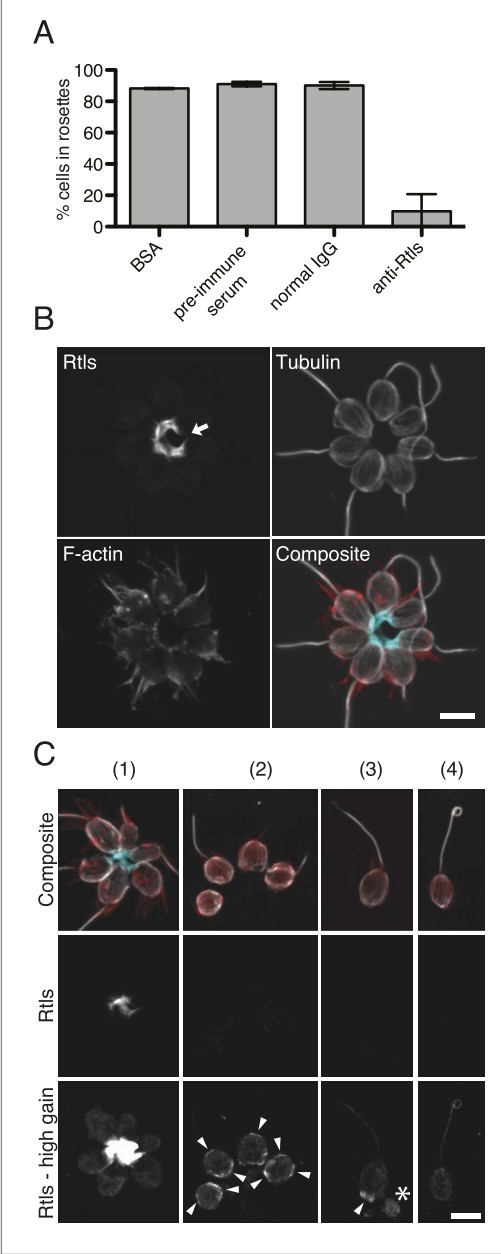

**Figure 6**. Rtls is required for rosette development and localizes to the center of rosettes. (**A**) Rosette development in wild-type *S. rosetta* was inhibited in the presence of 50 µg/ml anti-Rtls antibody, leading to a significant reduction in the percentage of cells in rosettes (one-tailed *t* test, p < 0.05) as compared to BSA, pre-immune serum, and IgG negative controls. Error bars show standard deviation. (**B**–**C**) The localization pattern of cell-associated Rtls differs between wild-type rosettes, chains, and single cells. (**B**) In rosettes, Rtls (cyan) was detected as a thick layer associated with the basal poles of the cells. Commonly, a gap was observed in the Rtls staining between one pair of neighboring cells in each rosette (arrow). The collar microvilli and filopodia were stained with

*Figure 6. Continued on next page*

Importantly, treatment of wild-type cells with 50 µg/ml anti-Rtls did not result in a loss of cell viability or reduction in cell growth (*Figure 6—figure supplement 1B*). Therefore, we conclude that the function of secreted Rtls is specific to and essential for rosette development.

## Rtls localizes to the center of rosettes

The connection between Rtls function and rosette development was also reflected in its differential localization in wild-type rosettes, chains, and single cells. In wild-type cells, Rtls was highly enriched in the extracellular matrix-filled center of rosettes, where it was observed in a thick layer underlying the basal poles of all cells in the rosette (*Figure 6B,C*). While Rtls staining sometimes connected all cells in the center of rosettes (*Figure 6C*), in most instances Rtls was observed to connect all but one pair of neighboring cells (*Figure 6B*). Because rosettes form through a process of incomplete cytokinesis (*Fairclough et al., 2010*; *Dayel et al., 2011*), this discontinuous Rtls staining may reflect the history of cell division during rosette formation, with discontinuities in its distribution indicating adjacent cells that were not sisters. We hypothesize that Rtls regulates rosette development by interacting with components of the extracellular matrix (ECM), which has previously been shown to fill the center of rosettes (*Dayel et al., 2011*).

In wild-type single cells and chain colonies, the subcellular localization and apparent abundance of Rtls were notably different than in rosettes. Despite the fact that equivalent levels of Rtls were detected in lysates from rosette-induced and -uninduced cultures (*Figure 5F*), little to no Rtls signal was detected by immunofluorescence when wild-type single cells and chains were imaged with the settings used for visualizing Rtls in rosettes (*Figure 6C*). Because Rtls has a predicted secretion signal, it is possible that *S. rosetta* chain colonies and single cells released Rtls into their aquatic environment, where it may have been washed away during processing of cells for immunofluorescence. After increasing the gain used during confocal imaging, we were able to detect Rtls in cell membrane-associated patches in wild-type single cells and chains, but these patches were absent from Rosetteless mutant cells (*Figure 6C*, *Figure 6—figure supplement 2*). The patches of Rtls localization were most often located near the basal pole of each cell, but were sometimes detected at the apical pole or along the lateral regions of the cell (*Figure 6C*). In no case was the Rtls staining in single cells or

*Figure 6. Continued*

phalloidin (red) and anti-tubulin staining (white) was used to highlight the cell body and flagellum. (**C**) Rtls localization in (1) wild-type rosettes, (2) wild-type chains, (3) wild-type single cells, and (4) Rosetteless mutant single cells. In single cells and chains imaged as in *Figure 6B* ('Rtls', laser intensity = 2.0, zoom = 2.5, gain = 544), Rtls signal was nearly undetectable. However, when imaged with a higher photomultiplier gain ('Rtls–high gain', laser intensity = 2.0, zoom = 2.5, gain = 750), Rtls was detected in membrane-associated patches (arrowheads) in wild-type single cells and chains, but not in Rosetteless cells. Wild-type single cells and chains frequently also had immunoreactive material deposited on the slide adjacent to the cells (asterisk). All cell types showed faint, diffuse fluorescence throughout the cell body, but this was likely the result of non-specific staining (*Figure 6—figure supplement 2*). Scale bars = 5 µm.

The following figure supplements are available for figure 6:

**Figure supplement 1**. Anti-Rtls blocks rosette development.

**Figure supplement 2**. Validation of the anti-Rtls antibody in immunofluorescence.

chains as intense as the Rtls staining observed in the cores of rosettes.

In summary, three findings demonstrate that Rtls function is necessary for and specific to rosette development: (1) the localization of Rtls protein is developmentally regulated and most abundant in the core of rosettes, (2) Rosetteless mutant cells fail to form rosettes but are otherwise wild type, and (3) secreted Rtls protein is essential for rosette development, while being dispensable during other stages of the *S. rosetta* life history.

## Discussion

The *rosetteless* gene is the only gene yet known to be required for choanoflagellate multicellular development. The molecular mechanisms by which Rtls regulates rosette formation remain unknown, but the developmentally regulated secretion of Rtls protein into the ECM-filled space in the center of rosettes (*Figure 6B*) likely provides some important clues. Rtls may stabilize the connections between rosette cells by interacting with the ECM in a manner akin to the lecticans, a family of animal C-type lectins that stabilize cartilage and other connective tissues by cross-linking carbohydrates and proteins in the ECM (*Ruoslahti, 1996*). Such a role would be consistent with the observation that the Rosetteless mutant produces wild-type chain colonies, as one of the main differences between rosettes and chain colonies lies in the stability and mechanical robustness of rosettes as compared to chain colonies. A second possible hint regarding Rtls function stems from the fact that the Rtls CTLDs most closely resemble animal CTLDs that preferentially bind mannose (*Drickamer, 1992*), such as mannose binding protein and pulmonary surfactant protein A, each of which functions in innate immunity as pattern recognition receptors (*Takahashi et al., 2006*). Because rosette development is regulated by bacterial signals (*Alegado et al., 2012*), Rtls may play a role in substrate recognition and cell signaling. Future work on the biochemical and physiological roles of Rtls will enable the discovery of other proteins in the rosette development regulatory pathway, while also potentially providing insights into the ancestral functions of CTLD-containing proteins.

The discovery that Rosetteless regulates rosette development provides a starting point for investigating the relationship between animal and choanoflagellate multicellularity. CTLD-containing proteins have previously been shown to regulate cell adhesion and development in animals (*Reidling et al., 2000*; *Iba et al., 2001*; *Kulkarni et al., 2008*; *Chin and Mlodzik, 2013*), offering intriguing parallels with the role of the Rosetteless CTLD protein in the control of rosette development. However, the molecular functions of Rtls are currently unknown and it is therefore unclear whether they are conserved in animal CTLD-containing proteins. Moreover, while the genomes of diverse animals encode Rtls-like proteins containing a signal peptide, two C-type lectin-like domains, and serine-threonine-rich low complexity regions (e.g., the placozoan *Trichoplax adhaerans* (XP_002112548), the cnidarian *Hydra vulgaris* (XP_002155329), the nematode *Caenorhabditis elegans* (NP_501369.1), and the fish *Danio rerio* (XP_005158004); relevant motifs detected by SMART, [*Letunic et al., 2012*]), it is not clear whether the similarities among these proteins and Rtls are the result of homology or convergent evolution (*Figure 5—figure supplement 2*). Although it is not straightforward to reconstruct the evolutionary relationships among *S. rosetta* and animal CTLD-proteins, the future analysis of additional rosette defect mutants promises to illuminate the remaining rosette regulatory pathway and reveal whether this pathway is conserved in the regulation of animal multicellularity.

Forward genetic screens have been vital tools for uncovering fundamental mechanisms driving development in eukaryotic model organisms, including *Saccharomyces cerevisiae*, *Drosophila melanogaster*, *C. elegans*, *Mus musculus*, *D. rerio*, and *Arabidopsis thaliana* (*Hartwell et al., 1970*; *Brenner, 1974*;

*Nüsslein-Volhard and Wieschaus, 1980*; *Mayer et al., 1991*; *Haffter et al., 1996*; *Kasarskis et al., 1998*), but such approaches have been restricted to a relatively small number of taxa that represent a small fraction of eukaryotic diversity (*Abzhanov et al., 2008*). Expanding the phylogenetic reach of forward genetic approaches will allow for a more rigorous and complete interrogation of the origin and evolution of animal development. The establishment of forward genetics in choanoflagellates has provided the first insights into the genetic underpinnings of development in these evolutionarily relevant organisms and promises to illuminate mechanisms underlying intercellular interactions in the progenitors of animals.

## Materials and methods

### Culture media

Unenriched artificial seawater (ASW), cereal grass media (CG media), and high nutrient (HN) media were prepared as described previously (*Levin and King, 2013*). HN media (250 mg/l peptone, 150 mg/l yeast extract, 150 µl/l glycerol in unenriched sea water) was made by diluting Sea Water Complete Media (*Atlas, 2004*) to 5% (vol/vol) in ASW. *A. machipongonensis* conditioned media (ACM) was made from the sterile supernatant of the liquid *A. machipongonensis* culture (ATCC BAA-2233 [*Alegado et al., 2013*]) grown shaking for 48 hr in HN media at 30°C to an $OD_{600}$ of 0.30–0.39 and filtered through a 0.2 µm filter to remove bacterial cells and detritus.

The above conditions were used for the isolation of all mutants except Rosetteless. For this mutant, ACM was prepared in CG media and was grown for 24 hr to an $OD_{600}$ of 0.1. Rosetteless clonal isolation steps used a mixture of 20% *Algoriphagus* conditioned CG media, 5% fresh CG media, and 75% ASW (vol/vol).

### Strains

The parental strain for the screen was SrEpac (ATCC PRA-390; accession number SRX365844), which contains *S. rosetta* grown in the presence of *Echinicola pacifica* bacteria (*Nedashkovskaya et al., 2006*; *Levin and King, 2013*), previously described as 'Isolate C' in *Levin and King (2013)*. SrEpac was generated through serial clonal isolation to ensure a genetically homogeneous background for the screen, and frozen stocks of SrEpac were thawed prior to each mutagenesis to limit the accumulation of random mutations. SrEpac cultures were haploid when passaged every 2–3 days in HN media (*Levin and King, 2013*). During each mutagenesis treatment, an SrEpac culture was divided into two; one half was mutagenized and the other half underwent all incubations, washes, and clonal isolation steps of the protocol except for the mutagenesis.

The Isolate B culture used in the cross (accession number SRX365839) contains *S. rosetta* grown in the presence of *A. machipongonensis* bacteria. Isolate B was diploid when passaged with scraping every 3 days in CG media (*Levin and King, 2013*).

### Mutagenesis

To determine a mutagen dose to be used in the screen, we titrated each mutagen over three orders of magnitude and examined the cell number of mutagenized vs unmutagenized cultures 24 hr later (*Figure 3—figure supplement 1A*). For both EMS and X-rays, we observed a general decrease in cell number following increased mutagen dose, suggesting that the mutagen was effective. For the screen, we used mutagen doses of 0.3% (vol/vol) EMS and 6300 rem of X-rays, as both treatments showed an intermediate effect on cell number, but this effect varied considerably among mutagenesis trials.

For EMS mutagenesis, approximately $10^6$ cells were washed and resuspended in 1 ml ASW. Liquid EMS (ethyl methanesulfonate, Sigma, St. Louis, MO) was added to 0.3% (vol/vol) and cells were incubated 1 hr at room temperature. The EMS was subsequently removed and neutralized by washing the cells three times in 5% sodium thiosulfate in ASW (wt/vol) before returning the cells to their initial media (HN or CG media) for 24 hr of recovery. In parallel with the isolation of the Rosetteless mutant, we also co-isolated a wild-type strain (C2E5) that underwent all washing and clonal isolation steps but was not mutagenized.

Fox X-ray mutagenesis, approximately $10^6$ cells were transferred into 35 mm diameter tissue culture dishes (Thomas Scientific, Swedesboro, NJ) and placed in an X-ray cabinet (Faxitron 43855C) 30.3 cm from the X-ray source with the lids of the dishes removed. Cultures were irradiated at the 125 V, 3 mA setting for 3 hr, which corresponded to a dose of approximately 6300 rems. Although we observed only mild choanoflagellate death from the mutagenesis treatments (*Figure 3—figure supplement 1A*), there was significant death and/or growth inhibition of the *E. pacifica* bacteria following X-ray

mutagenesis. Therefore, to avoid *S. rosetta* starvation, we added 500 µl of an unmutagenized, liquid culture of *E. pacifica* bacteria to the *S. rosetta* after X-ray mutagenesis and resuspended the cells in 10 ml HN media before a 24 hr recovery.

To measure the X-ray dose delivered under these conditions, we placed ring dosimeters at the same position and exposed them for 1 min, 1.5 min, or 1.75 min to generate a standard curve. By linear regression, we obtained the following formula with a fit of $R^2 = 0.997$: millirems of exposure = 35,091 * (minutes exposure)—4531.1. Given this equation, we calculated that the X-ray mutagenesis dose corresponded to approximately 6300 rems.

## Screen for rosette defect mutants

SrEpac cells were mutagenized either with 0.3% (vol/vol) liquid EMS (ethyl methanesulfonate) for 1 hr or exposed to 6300 rem of X-rays (*Figure 3—figure supplement 1A*). 24 hr after mutagenesis, control and mutant clones were isolated by dilution-to-extinction into 150 µl screen media (20% ACM, 40% HN media, 40% ASW [vol/vol]) in 96-well plates. Cells were plated at an approximate density of 1 cell/150 µl (i.e., 1 cell/well). The probability that each isolate underwent a clonal bottleneck during this step was 0.70 to 0.89, calculated using the Poisson distribution and the number of choanoflagellate-free wells per plate (*Levin and King, 2013*). After 5–7 days, clonal populations were visually screened for mutants deficient in rosette formation (*Figure 2*). Selected controls and rosette defect mutants were expanded in 3 ml 10% ACM in 6-well plates to verify the phenotype.

In total, we isolated 19 candidate mutants. Nine were eventually verified as rosette defect mutants through repeated re-isolation and testing of rosette induction. A tenth mutant had a mild growth defect and was thus discarded. Of the remaining candidate mutants, most were isolated as thecate cells, a cell type that is not competent to form rosettes (*Figure 1*) (*Dayel et al., 2011*), but upon further passaging the cells in these cultures began to form rosettes. We concluded that the rosette defect phenotypes initially detected in these clones were likely a result of epigenetic rather than genetic heritability, and we focused instead on the nine verified rosette defect mutants.

To ensure that each mutant and control isolate was truly clonal, a second clonal isolation step was performed into 96-well plates to an average of 1 cell/1500 µl (i.e., 1 cell/10 wells). The probability that each isolate underwent a clonal bottleneck during this step was 0.935 to 0.997, resulting in an overall probability of 0.991–0.999 that each isolate underwent a clonal bottleneck at least once.

## Quantification of mutant rosette defects (*Figure 3A* and *Figure 3—figure supplement 1C*)

*S. rosetta* cultures were exposed to either ACM or live colony-inducing bacteria. For the live bacteria treatments, *A. machipongonensis* liquid cultures were grown shaking in HN media at 30°C for 24 hr. To begin the induction, *S. rosetta* cells were diluted to $10^4$ cells/ml in 3 ml HN media with either 20% ACM or 4 µl/ml of liquid *A. machipongonensis* culture. 48 hr after induction, we pipetted the culture vigorously and repeatedly to break up chain colonies, concentrated the cells fivefold by centrifugation, fixed an aliquot of the culture with formaldehyde, and assessed rosette formation by counting on a hemacytometer. Thus, our operational definition for rosettes only included those rosettes that were robust to vigorous pipetting.

## Imaging mutant rosette phenotypes

For all experiments to visualize mutant rosette phenotypes, cells were plated at a density of $10^4$ cells/ml in 3 ml of either HN media or 10% ACM in HN media (vol/vol). Cultures were imaged 48 hr after induction. For all non-fluorescent images, cells were visualized live (*Figure 3B,C*, and *Figure 3—figure supplement 1E*).

For the high magnification DIC images (*Figure 3B,C*), 96-well µclear flat bottom plates (Greiner) were coated with 0.1 mg/ml poly-D-lysine (Sigma) for 5 min and allowed to air dry for 5 min before gently transferring 100 µl of culture to the well with a cut-off pipet tip. Cells were allowed to settle for 5 min and imaged live at 63× oil immersion with a Leica DMI6000B microscope equipped with a Leica X-Cite 120 camera.

To visualize low magnification fields of view of the mutants following pipetting (*Figure 3—figure supplement 1E*), cells were pipetted rigorously to break up chain colonies and concentrated 30–100-fold by centrifugation. 10 to 20 µl of concentrated cells were imaged live on a slide at 10× on a Leica DMIL LED inverted compound microscope with a Leica DFC 300FX camera.

For the confocal slices through rosette colonies (*Figure 3—figure supplement 1D*), sterile, 8-well µ-slides (Ibidi, Germany) were coated with 0.1 mg/ml poly-D-lysine (Sigma) for 5 min and allowed to air dry for 5 min before gently transferring 250 µl of culture to the well with a cut-off pipet tip. Cells were fluorescently stained with 1 µl of 2.5 µg/ml FM 1-43X dye (Molecular Probes, Eugene, OR), fixed with 1 µl 25% glutaraldehyde (Electron Microscopy Sciences, Hatfield, PA), allowed to settle for 5 min, and imaged at 63× using a Zeiss LSM 700 confocal microscope. Single confocal slices are shown.

### Imaging mutant chain phenotypes (*Figure 3—figure supplement 2*)

Because chain colonies break up upon pipetting and because some of the mutants formed chains with very large clusters of cells, we attempted to visualize the chain phenotypes while manipulating the cells as little as possible. Cells were diluted at a 1:10 ratio into 10 ml of HN media in 25 cm$^2$ culture flasks (Corning, NY). 24 hr later, we imaged the chain colonies at the bottom of the flask at 10× using a Leica DMIL LED inverted compound microscope with a Leica DFC 300FX camera. Images were manually false colored to highlight the chain colonies that were in focus.

### Genome sequencing

We sequenced the genomes of the Rosetteless mutant, the parental strain from which it was derived, and an unmutagenized wild-type strain (C2E5) that was isolated and cultured in parallel with Rosetteless. We prepared genomic DNA from mutant and wild-type *S. rosetta* cultures by phenol chloroform extraction and used a CsCl gradient to separate *S. rosetta* and *E. pacifica* DNA by GC content (*King et al., 2009*). Multiplexed, 100 bp paired-end libraries were sequenced on an Illumina HiSeq 2000. Raw reads were trimmed with TrimmomaticPE (*Lohse et al., 2013*) to remove low quality base calls. Trimmed reads were mapped to the *S. rosetta* reference genome (*Fairclough et al., 2013*) using Burrows-Wheeler Aligner (*Li and Durbin, 2009*), and we removed PCR duplicates with Picard (http://picard.sourceforge.net). Rosetteless was sequenced to a median coverage of 71× and over 93% of the reference genome had at least 10× coverage, while the parental strain and C2E5 were each sequenced to a median coverage of 50–60× and over 91% of the genome had at least 10× coverage. We realigned reads surrounding indel calls using GATK (*DePristo et al., 2011*) and called variants using SAMtools and bcftools (*Li et al., 2009*). To obtain the high quality variant calls (*Figure 4—figure supplement 1*), we removed all variants that were called with a quality score below 100 in addition to all variants that were called as heterozygous, since we expected these haploid genomes to yield homozygous calls. We focused on detecting single nucleotide variants (SNVs), because Rosetteless was isolated following EMS treatment.

### Identifying Rosetteless SNVs

After filtering the detected SNVs by quality score, we found that Rosetteless contained 25,160 high-quality SNVs, 25,143 of which (99.93%) were shared among Rosetteless and at least one of the wild-type strains (the parental strain and C2E5), meaning that they were segregating polymorphisms, which were unlikely to contribute to the Rosetteless phenotype. We experimentally validated all of the predicted Rosetteless-specific SNVs (*Figure 4—figure supplement 1B*). Short regions of genomic DNA flanking SNVs predicted to be unique to Rosetteless were amplified by PCR using a 1:1 mix of Taq (New England Biosciences, Ipswich, MA) and Pfu (Thermo Fisher Scientific, Waltham, MA), gel extracted using the GeneClean II kit (MP Biomedicals, Santa Ana, CA), and analyzed by Sanger sequencing. SNVs were considered 'verified' if they were present in Rosetteless gDNA but absent from gDNA from the parental strain. The supercontig 8 splice donor mutation (*rtls*[*l1*]) was the only Rosetteless-specific SNV predicted to alter a coding region, and we confirmed that this mutation was present in Rosetteless and absent from the parental strain by PCR and Sanger sequencing (*Figure 4—figure supplement 1C*). In contrast, when we attempted to verify the other 16 detected Rosetteless SNVs, only three were verified as polymorphic between Rosetteless and the parental strain. Of the remaining variants, three were false-positive variant calls in Rosetteless, two lay within regions of the reference genome that were misassembled, and eight were false-negative variant calls, where shared, segregating polymorphisms were not identified in the parental strain or the C2E5 wild-type strain. Thus despite the fact that the vast majority of called SNVs were high quality and independently called in all three samples, the enrichment of poor SNV calls in the Rosetteless-specific set meant that of the 17 potentially unique SNVs originally identified in the Rosetteless genome, there remained only four verified SNVs, including a predicted splice donor mutation at supercontig 8: position 427,804.

We were initially surprised to find such a small number of unique mutations in Rosetteless. However, it is possible that the EMS mutagenesis was ineffective prior to the isolation of Rosetteless, which is consistent with the fact that the Rosetteless mutagenesis did not result in substantial cell death (*Figure 3—figure supplement 1A*). Thus, despite the fact that Rosetteless was derived from a culture treated with EMS, it may in fact be a spontaneous mutant.

The raw reads for the SrEpac parental strain, the C2E5 wild-type co-isolate, and Rosetteless are publicly available (accession numbers SRX365844, SRX476076, and SRX476075, respectively). All alignments of protein sequences were made using fast statistical alignment (*Bradley et al., 2009*).

## Validating and genotyping additional SNVs

To investigate whether additional mutants isolated in this screen (i.e., mutant classes B–G; *Table 1*) bore mutations in *rtls*, we used Phusion polymerase (New England Biosciences) to amplify the coding region of the *rtls* from each mutant prior to cloning into the pCR 2.1 vector (Invitrogen). The coding region was divided into three regions for each mutant, using the following primer pairs: Rtls_L1/Rtls_R3, Rtls_L5/Rtls_R4, and Rtls_L3/Rtls_R2 (*Supplementary file 1*). The full insert of each clone was analyzed by Sanger sequencing. No mutations were found in *rtls* in any of the eight remaining rosette defect mutants.

To genotype microsatellites with size polymorphisms larger than 30 bp (e.g., the indel1 marker), we separated PCR products on a 2% agarose gel. To genotype smaller microsatellites, we fluorescently labeled PCR products (*Schuelke, 2000*) and analyzed the size polymorphisms by fragment analysis on a 3730XL DNA Analyzer (Applied Biosystems). The gt_indel_2 and gt_indel_7 primer sets included an M13 site on the left primer to enable fluorescent labeling in a 3-primer reaction, while the gt_indel_9 left primer was directly fluorescently labeled (*Supplementary file 1*).

## Performing a choanoflagellate cross

### Part 1: isolation of a haploid strain of isolate B

As Rosetteless and its SrEpac parental strain had very few genetic differences that could be tracked in a backcross (*Figure 4—figure supplement 1*), we opted instead to perform a cross between the Rosetteless mutant and Isolate B, which was previously sequenced (*Levin and King, 2013*) and was predicted to have 39,451 polymorphic markers relative to Rosetteless (*Figure 4—figure supplement 2*). Isolate B had been maintained as a diploid culture, which was not suitable for crossing to the haploid Rosetteless strain. However, as Isolate B exhibited genome-wide homozygosity, we reasoned that if we could induce Isolate B to undergo meiosis and generate an Isolate B haploid strain, this haploid strain would inherit the same predicted markers as the sequenced, diploid Isolate B strain. Thus, our first goal was to isolate a haploid strain from Isolate B.

Isolate B consists of *S. rosetta* cells that are fed *A. machipongonensis* bacteria and cultivated in CG media (*Levin and King, 2013*). We induced Isolate B to become haploid by passaging the culture with a 1:2 or 1:5 dilution every 2 to 3 days in CG media for several weeks. Although Isolate B is typically thecate when diploid, this passaging regime resulted in a culture consisting mostly of rosettes. We measured the ploidy of the culture by flow cytometry as in *Levin and King (2013)* and found that approximately 51% of the population was haploid. To establish a clonal, haploid line of Isolate B, we isolated cells by limiting dilution into 96-well plates containing 10% CG media in ASW. The probability of clonal isolation during this step was 0.93. We selected 12 isolates to expand into larger volumes and measured the ploidy of each clonal population by flow cytometry. We selected one isolate that consisted almost entirely of haploid cells to proceed.

### Part 2: induction of mating

Our next goal was to induce mating between Rosetteless and the haploid Isolate B culture. *S. rosetta* mating can be induced by transferring a stationary phase culture to nutrient poor media for several days (*Levin and King, 2013*). As *S. rosetta* can undergo both self-fertilization and outcrossed mating (*Levin and King, 2013*), we expected this procedure would generate a mixed population of cells that would include: (1) outcrossed heterozygous diploids containing both mutant and wild-type alleles, (2) homozygous diploids generated from the self-fertilization of either parental type, and (3) haploid, parental-type cells that never underwent mating. For the purposes of the cross, we were interested in only the outcrossed, heterozygous diploids. To enrich for these cells, we needed to both maximize the proportion of the population that was induced to mate and attempt to have an equal mix of Rosetteless and Isolate B cells present when mating occurred.

We first attempted to grow the Rosetteless and Isolate B cultures at similar rates through similar passaging regimes. The *S. rosetta* in Rosetteless and Isolate B are cultured with two different species of bacteria, so we added 1 ml of an *A. machipongonensis* liquid culture to Rosetteless and 1 ml of an *E. pacifica* liquid culture to Isolate B to ensure that both choanoflagellate cultures were fed to both bacterial species. The two isolates were passaged daily for 8 days to a starting cell density of $5 \times 10^4$ cells/ml in 10 ml CG media. This was continued for 8 days. On the first 2 days, 1 ml of liquid *E. pacifica* culture in HN media was added to each isolate to encourage rapid growth.

On the ninth day, we set up the starvation conditions for the cross. We mixed the cells together by adding $5 \times 10^5$ cells from each culture to 9 ml CG media and 1 ml of liquid *E. pacifica* culture. The next day, the mixed culture was pelleted and resuspended in 10 ml ASW to starve the cells and induce mating (*Levin and King, 2013*). After 11 days of starvation in ASW, we measured the ploidy of the culture and found that approximately 75% of the cell population had become diploid, suggesting that mating had occurred. 3 ml of the starved culture was then added into 10 ml CG media and cells were subsequently passaged every 1–3 days for 8 days to revive the cultures from their starved state.

## Part 3: identification of outcrossed diploids

Following mating, we isolated clones and identified cells that had undergone outcrossed mating through genotyping. We reasoned that genotyping each clone at three unlinked markers could provide evidence for meiosis through independent assortment, while also allowing for multiple, genetically distinct, progeny to be isolated from each flask. Because the chromosome number of *S. rosetta* has not been determined, we selected three markers, each on one of the three largest assembled supercontigs (2.5 Mb, 2.0 Mb, and 1.9 Mb in size), to help ensure that the markers would either be on different chromosomes or far enough apart to be unlinked.

We isolated clones by limiting dilution into twenty 96-well plates containing 10% CG media in ASW (vol/vol). The probability of clonal isolation in this step was 0.86. After 1 week of growth, 384 isolates were expanded into 4 ml 50% CG media in 6-well plates to accumulate enough biomass for genotyping. After 4–7 days growth, we pelleted 2 ml of the culture and extracted DNA using a base/Tris method as follows. We resuspended the pellet in 20 µl base solution (25 mM NaOH, 2 mM EDTA), transferred the sample into PCR plates, and boiled at 100°C for 20 min, followed by cooling at 4°C for 5 min. We then added 20 µl Tris solution (40 mM Tris–HCl, pH 7.5) and used 1 µl of this sample as the DNA template for each of our genotyping reactions.

To identify which clonal populations were the result of outcrossed mating (as opposed to self-mating), we genotyped each of the clonal isolates at three microsatellite markers that were polymorphic between the Rosetteless and Isolate B parental strains (gt_indel_2, gt_indel_7, and gt_indel_9; *Supplementary file 1*). Any clonal populations that had undergone outcrossed mating were expected to be heterozygous at all three markers, while those that did not mate or self-fertilized were expected to be homozygous at all three markers. Of the 384 genotyped clones, seven were heterozygous at all three markers, suggesting that these clones were the product of outcrossed mating. All remaining clones were homozygous at all three markers, but 372 clones were homozygous for the Isolate B alleles, while only three were homozygous for the Rosetteless alleles, raising the possibility that the Rosetteless mutant was less viable than Isolate B under the starvation conditions used to induce mating. Such differential viability may also explain the low rate of outcrossed mating, if few Rosetteless haploids survived to mate with Isolate B haploids. But despite the low frequency of outcrossed mating, we proceeded with the cross using the heterozygous diploids.

## Part 4: induction of meiosis and initial isolation of haploid, meiotic progeny

We next induced meiosis in the outcrossed heterozygotes to complete the sexual life cycle and obtain recombinant, haploid progeny from the cross. We expanded the heterozygotes in 10 ml CG media and measured the ploidy of the cultures, expecting that the heterozygotes would form largely diploid population of cells. In four of the seven cultures, the population remained mostly diploid, whereas there was a substantial haploid population in the remaining three cultures, suggesting the meiosis had already occurred for a large subset of these cells. To ensure that we could isolate the products of independent meioses, we divided the four, mostly diploid cultures into ten total flasks and then passaged these cultures rapidly to induce meiosis (*Levin and King, 2013*). 1 ml of liquid *E. pacifica* culture was added to each flask. The ten flasks were passaged every 1–2 days for 7 days, scraping to dislodge thecate cells and adding liquid *E. pacifica* each time.

We next repeated the clonal isolation and genotyping steps, as above, to identify the haploid products of meiosis. We genotyped 288 clonal populations and identified 32 isolates (11%) that were homozygous at all three genotyped markers. All of these putatively haploid cultures formed either rosettes or chain colonies. We also identified 17 clonal cultures that were homozygous at some markers but heterozygous at others; these apparently diploid clones were presumably generated from the products of meiosis that later underwent a second round of mating before clonal isolation. Notably, the majority of the isolates (83%) were cultures of thecate cells that were heterozygous diploids and did not undergo meiosis.

## Part 5: isolating additional haploid progeny

Based on the genotyping results from Part 4, we hypothesized that the differentiation into the thecate cell type was a morphological correlate of diploidy. Therefore, we thawed one of the original heterozygous diploid isolates and again attempted to induce meiosis, this time while excluding the thecate cells during passaging. After thawing the heterozygous isolate into 10% CG medium (vol/vol in ASW), we scraped the culture to dislodge thecate cells and immediately divided the culture into six flasks. We passaged each flask every 2 days without scraping to induce meiosis and select against the thecate cells. We then repeated the clonal isolation and genotyping steps as above for 154 clonal isolates, of which 150 were haploid progeny (97%). Of the 182 total haploid progeny, four were duplicates of other isolates (as they shared matching genotypes at >90% of markers), and these were excluded from further analysis. Therefore, the mapping cross described here allowed for the isolation of 178 independent haploid progeny.

## Rosetteless: dominant or recessive?

We sought to investigate whether the Rosetteless phenotype was dominant or recessive by examining the phenotypes of the heterozygotes isolated after mating Rosetteless to Isolate B. All seven heterozygous cultures contained predominantly thecate cells—a cell type that is not competent for rosette formation (*Figure 1*) (*Dayel et al., 2011*)—and three of these cultures also contained a small number of rosettes. The presence of rosettes (and absence of chain colonies) suggested that the Rosetteless phenotype might be recessive. However, culture conditions that favor rosette or chain colony development over the production of thecate cells (i.e., rapid passaging in nutrient-rich media) also favor meiosis, meaning that the rare rosette colonies could represent a minority of cells that had already undergone meiosis. Therefore, we were not confident that rosettes were developing from diploid cells and could not definitively determine whether the Rosetteless phenotype was dominant or recessive.

## Linkage map construction

From the genome sequences of Rosetteless and Isolate B, we identified 39,451 putative polymorphic positions that could be used as markers to genotype the mapping cross. We prioritized markers that were: (1) on supercontig 8 near to the 427,804 $rtls^{l1}$ splice donor position, to increase our confidence in the mapping; (2) validated Rosetteless-specific SNVs (*Figure 4—figure supplement 1B*), because these were other plausible candidates for causing the Rosetteless phenotype; and (3) markers located near the ends of supercontigs, as linkage detected between these positions and markers on other supercontigs would allow for an improved linkage map of the *S. rosetta* genome.

We genotyped 182 haploid isolates at 60 markers, for a total of 10,920 genotyping reactions. In addition to the three microsatellite markers, the cross progeny were further genotyped at 57 markers using KASP technology (LGC Genomics, Beverly, MA; *Figure 4—source data 1* and *Supplementary file 2*). We obtained genotype data for 91% of these reactions. Four of the cross isolates were duplicates of other isolates in the set and were excluded from further analysis. The program R/qtl (*Broman et al., 2003*) was used to construct a preliminary linkage map from the Rosetteless–Isolate B cross progeny. For each genotyping reaction, Rosetteless alleles were coded as 'A' and Isolate B alleles were coded as 'H' to emulate a backcross and allow for haploid genetics to be analyzed. We included in the linkage map the 60 markers that were genotyped in over 90 progeny and the data from the 175 nonduplicate individuals that were genotyped in at least 30 markers each. We constructed linkage groups using a maximum recombination fraction of 0.4 and a minimum LOD of 5 (i.e., 1 in 100,000 odds of linkage). Following linkage group construction by R/qtl, we used the physical linkage known from the *S. rosetta* genome assembly to manually link markers on the same supercontig into a single linkage group. The linkage map is shown in *Figure 4—figure supplement 3* and the genotype data and linkage group assignments for each marker are available in *Figure 4—source data 1*.

For 78% of the genotyped markers, we observed that each allele was present in approximately 50% of progeny, as expected from Mendel's Law of Segregation (*Mendel, 1866*). However, we observed some segregation distortion for 13 of the markers examined. Four of these markers had missing data for 50–80 of the genotyped progeny, suggesting that an error in calling one of the alleles may be responsible for the apparent distortion. We also observed segregation distortion for four of the super-contig 2 markers and the gt2 splice donor marker that were each genotyped in the majority of the haploid progeny. The Rosetteless phenotype and splice donor mutation were present in 82% of the isolated progeny (145/177), while the remaining progeny readily formed rosette colonies, (*Figure 4*). We believe this segregation distortion may be related to the Rosetteless phenotype itself. As chain colonies break up into many individual cells, it may be more likely that chain-forming cells will be selected during the clonal isolation process as compared to cells in rosettes. However, regardless of the cause of the segregation distortion, we concluded that the defect underlying the Rosetteless phenotype was tightly linked to the *rtls^{I1}* splice donor mutation.

## RT-PCR and cloning of *rtls^{I1}* splice isoforms

We isolated RNA from the SrEpac parental strain and Rosetteless using the RNAqueous kit (Life Technologies) and prepared cDNA using oligo dT primers and Superscript II Reverse Transcriptase (Invitrogen) according to the manufacturer's instructions. After first strand synthesis, we performed PCR using primers flanking the candidate splice donor mutation (Rtls_2L and Rtls_1R; *Supplementary file 1*) and cloned the resulting bands into the pCR 2.1 vector (Invitrogen). Splice isoforms were determined by Sanger sequencing the full insert of each clone. From Rosetteless, we sequenced 11 clones that had retained intron 7 (isoform ii), 3 clones that had a late splice donor (isoform iii), and 21 clones that had an early splice donor (isoform iv), but no clones with the wild-type isoform. We repeated the above procedure using a primer specific to the wild-type exon 7/8 boundary (Rtls_5L and Rtls_1R) and obtained a single band from both wild-type and Rosetteless cDNA, each of which corresponded to the wild-type product upon sequencing. From both wild-type and Rosetteless samples, no bands were observed from the negative control PCRs, which used RNA that was not reverse transcribed as a template.

## Generation of the anti-Rtls antibody and purification of the recombinant Rtls epitope

The anti-Rtls antibody was generated using Genomic Antibody Technology (SDIX, Newark DE); in which rabbits were immunized with a DNA construct corresponding to this epitope: SSTPQQFPALV LEFPTPISESDVPAIELLLQSAGLPSNNPTGSSITVQLLSSQLVYIQLAGNFEQYAGELALKALN DQLIWQAGIPIAYVPLTSVLDQIQAT. The epitope is unique to Rtls and bears no resemblance to other polypeptide sequences in *S. rosetta*; when the amino acid sequence of the epitope was used to search the full catalog of *S. rosetta* proteins (using blastp), no other protein hit the epitope with an e-value less than 20 (i.e., the other hits were not statistically significant). The antibody was affinity purified against recombinant Rtls generated by SDIX.

Separately, we cloned, expressed, and purified recombinant protein corresponding to the epitope from wild-type *S. rosetta* cDNA that was prepared with oligo dT primers as described above. We amplified the epitope using Pfu (Finnzymes) and primers Rtls_epit_L1 and Rtls_epit_R1 (*Supplementary file 1*) and cloned it into the pGEX-6P GST-fusion expression vector. Protein was expressed in BL21 *E. coli* grown overnight at 16°C, purified using glutathione Sepharose 4B beads (GE Life Sciences, Pittsburgh, PA), and eluted with 50–100 mM glutathione. Elutions from multiple experiments were pooled and concentrated using a 30K Amicon Ultra-4 filter and the buffer was exchanged for the following protein wash buffer (50 mM Tris pH 7.4, 150 mM NaCl, 1 mM EDTA, 1 mM DTT, 0.01% Triton X-100). To visualize the purity of the recombinant Rtls epitope, 100 ng of protein was run on a 4–12% gradient SDS-PAGE gel (Bio-Rad, Hercules, CA) and silver stained with Fermentas PageSilver Silver Staining Kit (K0681) according to manufacturer's instructions. For western blot analysis, samples were transferred to PVDF membrane (Immobilon-FL, IPFL00010), blocked with Odyssey Blocking Buffer (Licor, 927–40,010), and probed with anti-Rtls antibody at 1:2500 followed by the secondary antibody anti-rabbit IRDye 800CW (Licor, 926–322111). The blot was imaged with the Licor Odyssey infrared imaging system.

## Dot blots

Wild-type and Rosetteless cultures were grown for 24 hr in the presence or absence of live *A. machipongonensis* bacteria. Following filtration through a 40 µm filter to remove bacterial biofilms, $1.5 \times 10^6$ cells from each culture were pelleted, resuspended in 10 µl of lysis buffer (10 mM Tris HCl

pH 8, 0.1 mM EDTA pH 8, 0.5% SDS wt/vol), and spotted directly onto a nitrocellulose membrane (NitroBind, GE Osmonics, Minnetonka, MN). Spots were allowed to air dry completely before blocking in 5% milk and treatment with anti-Rtls primary antibody (1:500). Primary antibody signal was detected using an IR-dye-conjugated secondary antibody (Licor Biosciences anti-rabbit 800 nm 1:10,000) and the Odyssey Infrared Imaging System (Licor Biosciences, Lincoln, NE). To test the specificity of anti-Rtls to Rtls protein on dot blot, the primary antibody was pre-incubated with a 100-fold molar excess of purified epitope at room temperature for 1 hr before application of the primary antibody to the membrane. Images were analyzed in ImageJ (*Schneider et al., 2012*) as follows: a box of constant area was placed over each dot to measure the integrated density of the area. The integrated density value for the secondary only control was subtracted from each sample to eliminate the signal due to mild autofluorescence of the membrane, and then each sample was normalized to the wild-type dot. Only dots processed on the same membrane were normalized in this manner.

### Blocking rosette formation with the anti-rtls antibody (*Figure 6A* and *Figure 6—figure supplement 1*)

To induce rosette development, HN media was inoculated with a single colony of live *A. machipongonensis*, vortexed, and aliquoted into a 96-well plate. Anti-Rtls antibody (1.25 mg/ml stock in PBS) was added to a final concentration of: 50, 25, 12.5, 6.25, 3.13, or 1.56 µg/ml and SrEpac wild-type cells were added to each well at a 1:5 dilution. Cells were incubated at room temperature in 100 µl total volume in a 96-well plate for 24 hr to induce rosette development, at which point cultures were vigorously pipetted and counted on a hemacytometer. Three negative control treatments were analyzed in parallel: (1) 50 µg/ml BSA (from a 1.25 mg/ml stock in PBS); (2) 14 µl of pre-immune serum (equivalent to the volume of antibody added in the 50 µg/ml condition); and (3) 50 µg/ml of a control, rabbit polyclonal antibody (from a 1.25 mg/ml stock in PBS, #A01008, Genscript, Piscataway, NJ). All conditions were tested in triplicate.

### Immunofluorescence microscopy

Live cells were allowed to settle for 30–60 min onto poly-L-lysine coated coverslips (BD Biosciences) and fixed in two steps: 5 min in 6% acetone followed by 10–15 min in 4% formaldehyde. Cells were stained with the anti-Rtls genomic antibody at 6.25 ng/µl (1:200), E7 anti-tubulin antibody (1:1000; Developmental Studies Hybridoma Bank), Alexa fluor 488 anti-rabbit and Alexa fluor 647 anti-mouse secondary antibodies (1:400 each; Molecular Probes), and 6 U/ml rhodamine phalloidin (Molecular Probes) before mounting in Prolong Gold antifade reagent with DAPI (Molecular Probes). To test the specificity of the antibody staining, 1 µl of anti-Rtls primary antibody was diluted in 190 µl block (1% BSA and 0.3% Triton X-100 in 100 mM PIPES pH 6.9, 1 mM EGTA, and 0.1 mM MgSO$_4$) and incubated with 9 µl of either protein wash buffer (see above) or purified epitope (equivalent to approximately 18 µg or a 60-fold molar excess) for 1 hr at room temperature before application of the primary antibody to the cells. Cells were imaged at 63× using a Zeiss LSM 700 confocal microscope (laser intensity = 2.0, zoom = 2.5, Rtls low exposure gain = 544, Rtls high exposure gain = 750).

### Guidelines for choanoflagellate gene naming

To date, no official rules have been established for naming choanoflagellate mutants or genes. Thus, we outline here proposed guidelines for naming choanoflagellate genes. Any genes with clear homology to named genes in other organisms should be referred to by the pre-existing name (e.g., *hsp90*). Any genes without clear homology to named genes should be given names that allude to the gene's function or the phenotype of the first mutant allele isolated. Before mutant genes are cloned, each mutant is given its own name, but renaming may be necessary once the causative mutation is identified. Gene names should be written in lower case and in italics. Specific mutations should be named with one letter for the last name of the first author of the publication describing the allele and one number for the order of allele isolated. Mutations are presented as an italicized superscript (e.g., *rtls*[*l1*]). Mutant names and protein names are written in upper case and non-italics. Since many three-letter abbreviations have already been used for genes in other organisms, we propose the use of four-letter abbreviations.

## Acknowledgements

We thank K Drickamer, I Hariharan, J Rine, R Alegado, A Glazer, J Bosch, S Nichols, J Nelson, and members of the King laboratory for discussions and comments on the manuscript, J Carlisle for help with screening, and E Waters for help with molecular cloning. TL was supported by a Graduate

Research Fellowship from the National Science Foundation (DGE 1106400). NK is an Investigator in the Howard Hughes Medical Institute and a Senior Scholar in the Integrated Microbial Biodiversity Program of the Canadian Institute for Advanced Research. This work was supported by funding from NIH NIGMS GM089977 to NK Genome sequencing reads have been deposited in the NCBI Short Read Archive with the following accession numbers: SRX365844, SRX476076, and SRX476075. The authors declare no competing financial interests.

## Additional information

### Funding

| Funder | Grant reference number | Author |
|---|---|---|
| Howard Hughes Medical Institute | | Nicole King |
| National Institutes of Health | GM089977 | Nicole King |
| Canadian Institute for Advanced Research | | Nicole King |

The funders had no role in study design, data collection and interpretation, or the decision to submit the work for publication.

### Author contributions

TCL, Conception and design, Acquisition of data, Analysis and interpretation of data, Drafting or revising the article; AJG, LW, Acquisition of data, Analysis and interpretation of data; NK, Conception and design, Analysis and interpretation of data, Drafting or revising the article

### Author ORCIDs

Tera C Levin, http://orcid.org/0000-0001-7883-8522

## Additional files

### Supplementary files

• Supplementary file 1. Primers used for genotyping and assessing splicing.

• Supplementary file 2. Polymorphic sequences targeted by KASP genotyping.

### Major dataset

The following datasets were generated:

| Author(s) | Year | Dataset title | Dataset ID and/or URL | Database, license, and accessibility information |
|---|---|---|---|---|
| Levin TC, Greaney AJ, King N | 2014 | C2E5 wild type isolate gDNA sequencing | SRX476076; http://www.ncbi.nlm.nih.gov/sra/SRX476076 | NCBI Short Read Archive. |
| Levin TC, Greaney AJ, King N | 2014 | Rosetteless mutant gDNA sequencing | SRX476075; http://www.ncbi.nlm.nih.gov/sra/SRX476075 | Publicly available at the NCBI Short Read Archive (http://www.ncbi.nlm.nih.gov/sra). |

The following previously published datasets were used:

| Author(s) | Year | Dataset title | Dataset ID and/or URL | Database, license, and accessibility information |
|---|---|---|---|---|
| Levin TC and King N | 2013 | Discovery of sexual reproduction in S. rosetta, isolate C (SrEpac) | SRX365844; http://www.ncbi.nlm.nih.gov/sra/SRX365844 | Publicly available at the NCBI Short Read Archive (http://www.ncbi.nlm.nih.gov/sra). |

| Levin TC and King N | 2013 | Discovery of sexual reproduction in S. rosetta, isolate B | SRX365839; http://www.ncbi.nlm.nih.gov/sra/SRX365839 | Publicly available at the NCBI Short Read Archive (http://www.ncbi.nlm.nih.gov/sra). |
|---|---|---|---|---|

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
