## [Decision Letter]

Thank you for sending your work entitled “A C-type lectin-like protein is required for development in the choanoflagellate *S. rosetta*” for consideration at *eLife*. Your article has been favorably evaluated by Janet Rossant (Senior editor), a Reviewing editor, and 3 reviewers.

The Reviewing editor and the reviewers discussed their comments before we reached this decision, and the Reviewing editor has assembled the following comments to help you prepare a revised submission.

This article by Levin et al., reports the first application of genetics to the model choanoflagellate species *Salpingoeca rosetta*, a pre-metazoan model for animal development. This organism develops from a single cell form into a multi-cellular form (rosettes) in response to environmental cues from its prey bacterium. The authors recently discovered and now take advantage of an extant sexual cycle to conduct chemical and X-ray mutagenesis. Nine mutants were uncovered through painstaking ocular inspection. One, Rosetteless, was characterized in detail here. The authors took multiple approaches to define and characterize the causative mutation, including genetic linkage analysis, whole genome sequencing, and characterization of the impact of the mutation on splicing. A genetic linkage map was constructed for the first time and used to show that the mutation segregates as a single mendelian trait on supercontig 8. The authors show that the mutation alters splicing of a putative C-type lectin, and make the case that this predicted cell surface protein is the root of the genetic defect. These studies represent a tour de force development and application of genetics to a powerful pre-metazoan model system.

The general consensus is that your manuscript represents landmark work as it describes the use of genetics for the first time in this group of organisms, and provides key insights into the evolution of animal multicellularity. In essence, this manuscript reports both significant conceptual and technical advances consistent with the high standards that one has come to expect from an *eLife* paper. We recommend that the paper be accepted for publication, provided that the following concerns are satisfactorily addressed.

1) Demonstrate antibody specificity: As the *rtls* mutation cannot be complemented, the specificity of the Rtls antibody is fundamental to the arguments of this paper. Presently, antibody specificity has not been adequately demonstrated. For example, three key pieces of data rely on antibody specificity; Rtls antibody inhibition experiment, Rtls quantification in WT +/- bacteria media, and protein quantification in *rtls* mutants.

While the data are compelling that the antibody is recognizing the protein, one could take a skeptical view that since a Western blot is not shown recognizing a single protein of the correct mass in wild-type, and missing or truncated in mutants, that they may have simply generated an antibody that happens to bind the outside of Choanoflagellates inhibiting their cell-cell interaction, but not due to inhibiting the activity of Rtls. Demonstrating antibody specificity by Western blotting is required and is missing.

Related to this issue is the use of dot-blots for quantification. While these assays can be informative, they have significant limitations, especially regarding epiptope availability, uneven antibody adhesion, and uneven binding to the substrate. As such, they are generally not regarded as the best way to quantify the amount of protein in a sample. Rather, the standard is to perform Western blot analysis and use densitometry to quantify the amount of protein in the desired band. In combination with the lack of antibody specificity, the authors cannot be sure what the differences in intensity in the dot blots are the result of.

This manifests in the curious results of Figure 5, which appears to have Rtls protein. However, the authors report earlier that they cannot detect appreciable amounts of properly spliced Rtls mRNA. However, Figure 5 shows that *rtls* mutants have ∼40% of the normal amount of protein. Is this due to truncated protein? Is their antibody not specific? Without a Western blot it is impossible to know what is actually being quantified. The authors do demonstrate that recombinant protein competes the antibody signal, but this raises a further question: what is the purity of the recombinant protein? A simple silver stained gel of the protein in the supplement would be informative, and again is typically reported when someone is reporting on the use of recombinant proteins.

In order to do this experiment correctly, Figure 5—figure supplement 1 should be western blots with recombinant protein, whole cell lysates from WT, WT Alg, *rtls* and *rtls* + Alg. Parallel blots should be probed with both the pre-immune and immune sera.

2) Homology of Rtls to metazoan proteins: The reviewers wish to understand whether Rtls is a multicellularity gene, or whether it is specific for rosette development. Although knowledge of the homology or Rtls to metazoan protein is not absolutely necessary and will probably not change the current conclusions, the relative importance of the finding reported (the mutant *rtls*) changes depending on whether Rtls is homologous or not to animal CTLD-containing proteins. Thus, if Rtls happens to be homologous to animal C-type lectins, then this will be a clear example of co-option (of adhesion proteins/domains) taken place at the origin of the metazoan lineage. If it is not, the finding is mostly important to understand the specific development of choanoflagellate colonies. Either way the finding is important enough, but the wider implications change. Thus, this issue should be clarified in order for the reader to fully grasp the implications of the findings here reported. One option is to perform phylogenetic trees to see whether Rtls (or the Rtls CTLDs) is/are homologous to metazoan C-type lectin containing genes. If trees are not possible, maybe the authors could try blast similarity network analyses. At a minimum, the authors should consider expanding the discussion to include: 1) whether *S. rosetta* or other choanoflagellates have other types of C-type lectins, 2) whether Rtls is also present in the non-colonial choanoflagellate *M. brevicollis* (from which genome is available), 3) whether CTLDs are specific to animals and choanoflagellates or are they present in other more ancient eukaryotes.

---

## [Author Response]

1) Demonstrate antibody specificity […]

2) Homology of Rtls to metazoan proteins […]

We start by addressing the generation and specificity of the Rosetteless antibody. As requested, we have revised the text (described in detail below) to more clearly explain the nature of the Rosetteless antibody and the evidence for its specificity. In addition, we have added two new pieces of data: (1) a silver-stained gel documenting the purity of the recombinant Rosetteless used to compete with endogenous Rosetteless in tests of antibody specificity and (2) a Western showing that Rosetteless antibody recognizes the recombinant protein (Figure 5—figure supplement 1).

First, we wish to clarify some information on the design and generation of the antibody. The Rosetteless antibody was raised against an epitope that we predicted should be present in all wild type and mutant Rtls isoforms (Figure 5; orange bracket). Therefore, our measurements of protein abundance (Figure 5) are expected to recognize both wild type and truncated versions of Rtls. To clarify this section of the Results, we have added the following text:

“To investigate endogenous Rtls protein in *S. rosetta* cells, we generated an antibody against residues 438–539, a region of the protein that is unique to Rtls and was expected to be present in all wild type and mutant Rtls isoforms (Figure 5, Methods, Figure 5—figure supplement 1). Using this antibody, we found that total Rtls protein levels in mutant cells were ∼25% that of wild type cells (Figure 5, Figure 5—figure supplement 1).”

We have also added the following section on Rtls epitope design to the Methods and provide the BLAST results below:

“The epitope is unique to Rtls and bears no resemblance to other predicted polypeptide sequences in *S. rosetta*; when the amino acid sequence of the epitope was used to search the full catalog of predicted *S. rosetta* proteins (using blastp), no other protein hit the epitope with an e-value less than 20 (i.e. the other hits were not statistically significant)”.TargetScore (Bits)Expect*S. rosetta*: PTSG_03555.1: Rosetteless214.1570.0*S. rosetta*: PTSG_04911.1: hypothetical protein25.79422.958*S. rosetta*: PTSG_08579.1: hypothetical protein25.79427.127*S. rosetta*: PTSG_06697.1: hypothetical protein25.40934.553*S. rosetta*: PTSG_11988.1: hypothetical protein25.02446.272

Unlike conventional antibodies, the anti-Rtls antibody was generated as a genomic antibody, using a process that involves the introduction into rabbits of a DNA construct encoding the epitope, rather than the injection of purified protein ([Brown, 2011]; http://www.sdix.com/Products/Custom-Antibody-Services/Antibody-Development/Genomic-Antibody-Technology.aspx). This means that there was no possibility of inadvertently introducing additional bacterial or choanoflagellate epitopes during antibody generation. After the rabbit antiserum was harvested, the antibody was also affinity purified and analyzed by ELISA by SDIX.

We now turn to a discussion of the evidence for the specificity of the Rosetteless antibody:

One of the gold standard methods for establishing antibody specificity is to show that the signal is present in wild type samples, but absent or altered in mutant or knock out samples. In the case of Rosetteless, the staining patterns detected by immunofluorescence microscopy in mutant and wild type cells were strikingly different (Figure 6), as were the levels of protein detected by dot blot (Figure 5). To summarize the main text, Rtls mutants are missing both the bright staining at the center of rosettes (because the mutants do not produce rosettes) and the fainter, cell-membrane associated patches found in wild type single cells and chains. By dot blot, the anti-Rtls signal was reduced in mutant cells (Figure 5, Figure 5—figure supplement 1) even though the antibody should recognize both wild type and truncated Rtls isoforms. These differences between mutant and wild type detected by immunofluorescence microscopy and dot blot are most consistent with the interpretation that the majority of the antibody signal recognizes an epitope that is altered between mutant and wild type cells.

As an additional test of the specificity of the antibody, we used recombinant protein as a competitor during immunofluorescence and in dot blots. As requested by the reviewers, we have added documentation of the purity of the recombinant protein (by silver stain) and a Western blot of the recombinant protein as recognized by anti-Rtls to the manuscript (Figure 5—figure supplement 1). The epitope has a predicted size of 38 kDa, and in our purifications it consistently appears at this size as a doublet, suggesting that some cleavage of the recombinant protein occurs in the bacteria. We indeed see this doublet as the predominant set of bands in the silver-stained recombinant protein sample and the only clear signal by Western blot.

In cells stained with the Rosetteless antibody, we see three main areas of immunofluorescence: (1) an extracellular meshwork in the center of rosettes, (2) in single cells and chains, a thin patch of membrane-associated staining that is most often found at the base of cells but sometimes seen in other parts of the cell, and (3) faint staining throughout the cell body in single cells and rosettes. If the Rosetteless antibody is pre-incubated with recombinant Rosetteless protein, the first two staining patterns (i.e. extracellular meshwork in center of rosettes and thin membrane-associated patches) disappear, presumably because those patterns result from antibody binding that specifically recognizes Rosetteless protein. The third pattern (i.e. faint staining throughout the cell body) persists following pre-incubation of the Rosetteless; we interpret this third pattern then to reflect either autofluorescence or non-specific staining. We are careful in the text to delineate that portion of the staining pattern that is specific to Rosetteless vs. that which reflects either autofluorescence or non-specific staining (in the legend for Figure 6—figure supplement 2): “In all cells following competition with the epitope there remained faint staining in the cell body; we thus infer that this is non-specific staining that does not reflect the distribution of Rtls protein in the cells”).

Finally, we agree with the reviewers that a Western of *S. rosetta* cell lysates would be a valuable contribution to our efforts to validate the Rosetteless antibody. Indeed, we have been trying to establish Rosetteless Westerns on *S. rosetta* cell lysates for the past 5 months. However, likely due to the nature of Rosetteless protein itself, our attempts at these Westerns have failed.

We find either that there is no signal or that the primary signal detected by the anti-Rtls antibody by Western blot is of a very high molecular weight protein that does not enter the gel. We believe that the explanation for this phenomenon lies in the predicted mucin-like domains of the Rtls protein; in similar S/T-rich regions, mucins are heavily glycosylated, making these proteins sticky and quite large [Harrop et al, 2011]. Indeed, because such ECM proteins may be decorated with a variable number of sugars and often undergo other forms of posttranslational processing (e.g. cleavage into active forms), a single band on a Western may not even be expected for this type of protein.

Still, we have made a number of attempts to modify our Western blotting protocol to accommodate heavily glycosylated proteins and solubilize the mucous-like *S. rosetta* lysate. These attempts include: 1) various cell lysis and solubilization methods; 2) syringing and other means of physical shearing; 3) pre-incubation of cell lysates with deglycosylases; 4) resuspension of lysate in 8M urea; and 5) the use of low-percentage acrylamide and acrylamide/agarose gels. However, in all attempts to date, we observe either a high molecular weight signal that does not enter the gel, or no signal at all.

Due to the challenges in working with fully glycosylated mucins, the use of dot blots for semi-quantitative analyses of mucin levels is not unusual [Harrop et al, 2011, Thomsson et al 2011]. Therefore, because Western analysis of Rosetteless in *S. rosetta* cell lysates has not yet been possible, we used dot blot analysis to measure the relative abundance of Rosetteless protein in wild type and mutant cells [Harrop et al, 2011, Thomsson et al 2011]. As the reviewers pointed out, dot blots are not ideal, as the high concentration of protein can lead to non-specific binding of antibodies and, because the proteins are not separated by size, it is not possible to see whether the source of the signal is from one protein or multiple proteins. Nonetheless, for proteins that do not properly enter into gels, it can provide a means by which to estimate their abundance. Similar to our procedure for validating the antibody by immunofluorescence, we used the pre-incubation of the recombinant epitope with anti-Rtls antibody to compete away the staining that specifically recognized the Rosetteless protein. In Figure 5—figure supplement 1, we demonstrate that this pre-incubation results in the loss of the majority, but not all, of the signal by dot blot. We also showed the residual signal following competition in the main figure (Figure 5), and explained in the figure legend for this panel that it was only a “semi-quantitative” method, thereby acknowledging the limitations with dot blots.

In summary, although we have been unable to perform a Western analysis on *S. rosetta* cell lysates, we have a number of independent observations and experiments that are all consistent with the interpretation that the anti-Rtls antibody displays a high degree of specificity for Rtls protein.

We now discuss the relationship between Rosetteless protein and CTLD-containing proteins in animals:

We have generated a new figure, Figure 5—figure supplement 2, to illustrate the diversity of CTLD-containing proteins in choanoflagellates. This figure shows the full domain architecture of all CTLD-containing proteins in the *S. rosetta* and *M. brevicollis* genomes, with similar animal CTLD domain architectures for comparison.

While Rtls may indeed be a multicellularity gene, we have not yet demonstrated this to be the case. We refrain from making this assertion because it is not currently possible to clearly assess the homology between Rtls and animal lectins. Specifically, the divergence time between animals and choanoflagellates means that orthology assignments often must be made on the basis of domain architecture and organization instead of amino acid alignments. This analysis cannot be convincingly carried out for Rtls, as its simple domain architecture is not sufficiently diagnostic. Finally, within animals, C-type lectins are a family of rapidly evolving genes that exhibit extensive duplications and rearrangements among taxa (e.g. [Sattler et al, 2012; Drickamer et al 1999]). It is thus not currently possible to assign clear orthology relationships among many of the animal C-type lectins, much less between animal C-type lectins and more evolutionarily distant CTLD-containing proteins found in choanoflagellates and other eukaryotes (e.g. [Wheeler et al, 2008]). We discuss the limitations in identifying homologous C-type lectins in the legend for Figure 5—figure supplement 2.